# The bHLH Transcription Factor OsbHLH057 Regulates Iron Homeostasis in Rice

**DOI:** 10.3390/ijms232314869

**Published:** 2022-11-28

**Authors:** Wujian Wang, Kamran Iqbal Shinwari, Hao Zhang, Hui Zhang, Lv Dong, Fengyu He, Luqing Zheng

**Affiliations:** College of Life Sciences, Nanjing Agricultural University, Nanjing 210095, China

**Keywords:** rice, bHLH transcription factor, *osbhlh057* mutants, Fe deficiency, OE lines, gene expression

## Abstract

Many basic Helix-Loop-Helix (bHLH) transcription factors precisely regulate the expression of Fe uptake and translocation genes to control iron (Fe) homeostasis, as both Fe deficiency and toxicity impair plant growth and development. In rice, three clade IVc bHLH transcription factors have been characterised as positively regulating Fe-deficiency response genes. However, the function of OsbHLH057, another clade IVc bHLH transcription factor, in regulating Fe homeostasis is unknown. Here, we report that OsbHLH057 is involved in regulating Fe homeostasis in rice. *OsbHLH057* was highly expressed in the leaf blades and lowly expressed in the roots; it was mainly expressed in the stele and highly expressed in the lateral roots. In addition, *OsbHLH057* was slightly induced by Fe deficiency in the shoots on the first day but was not affected by Fe availability in the roots. OsbHLH057 localised in the nucleus exhibited transcriptional activation activity. Under Fe-sufficient conditions, *OsbHLH057* knockout or overexpression lines increased or decreased the shoot Fe concentration and the expression of several Fe homeostasis-related genes, respectively. Under Fe-deficient conditions, plants with an *OsbHLH057* mutation showed susceptibility to Fe deficiency and accumulated lower Fe concentrations in the shoot compared with the wild type. Unexpectedly, the *OsbHLH057*-overexpressing lines had reduced tolerance to Fe deficiency. These results indicate that OsbHLH057 plays a positive role in regulating Fe homeostasis, at least under Fe-sufficient conditions.

## 1. Introduction

Iron (Fe) is an essential mineral nutrient for plant growth and development as it is responsible for numerous redox and electron transfer reactions, including chlorophyll synthesis and photosynthesis. Although abundant in the earth’s crust, Fe precipitates as insoluble Fe(III) oxides and hydroxides in aerobic or alkaline soils, making Fe unavailable to plants [1,2]. Fe deficiency stress results in leaf chlorosis and growth reduction and hence becomes a limiting factor for crop production and quality [3]. Excess Fe in plant cells, instead, could generate hydroxyl radicals which are toxic to cells via the Fenton reaction, leading to retarded growth [4,5]. Meanwhile, one billion people suffer from Fe deficiency anemia, particularly those who rely on plants for dietary Fe [6]. Thus, discovering sophisticated mechanisms by which plants control Fe homeostasis may profoundly impact crop yield and human nutrition.

Plants use two distinct Fe-uptake strategies, namely the reduction strategy (Strategy I) and the chelation strategy (Strategy II), which are employed by non-graminaceous species and graminaceous species, respectively [7]. In Strategy I plants, such as *Arabidopsis* (*Arabidopsis thaliana*), the solubility and mobility of insoluble Fe^3+^ in the rhizosphere are firstly improved by H^+^-ATPase AHA2 pumping protons to lower the pH of rhizosphere and PLEIOTROPIC DRUG RESISTANCE 9/ATP-BINDING CASSETTE G37 (PDR9/ABCG37) secreting coumarins. Then, FERRIC REDUCTASE OXIDASE 2 (FRO2) reduces the Fe^3+^ at the root cell surface to Fe^2+^, which finally is absorbed by IRON REGULATED TRANSPORTER 1 (IRT1) [8,9,10,11]. In Strategy II plants, such as rice (*Oryza sativa* L.), TRANSPORTER OF MUGINEIC ACID FAMILY PHYTOSIDEROPHORES 1 (TOM1) secretes 2′-deoxymugineic acid (DMA) to chelate Fe^3+^, thus enhancing the solubility of Fe^3+^ [12]. Then, the Fe^3+^-DMA complex is taken up by YELLOW STRIPE 1-LIKE 15 (OsYSL15) [13]. Rice (*Oryza sativa* L.) can also directly acquire Fe^2+^ via OsIRT1 [14,15].

For the adaptive fluctuation of Fe availability, plants have evolved a sophisticated regulatory mechanism of Fe homeostasis in which conserved basic Helix-Loop-Helix (bHLH) transcription factors (TFs) play a predominant role [16,17]. Studies in the model plants rice and *Arabidopsis* have revealed that the Ib, IIIa, IVb, and IVc clades of bHLH TFs form a precise regulatory network that contains two interconnected regulatory modules [17]. The first module acts upstream of the Fe uptake and transport genes. Clade Ib bHLH, bHLH38, bHLH39, bHLH100, and bHLH101 form heterodimers with clade IIIa bHLH TF FIT/bHLH29 to directly control the uptake of Fe [18,19,20,21]. OsIRO2, an Ib TF, can also interact with a clade IIIa bHLH TF *Oryza sativa* FER-LIKE FE DEFICIENCY-INDUCED TRANSCRIPTION FACTOR (OsFIT)/OsbHLH156. The complex of OsFIT/OsbHLH156 and OsIRO2 directly controls the expression of Strategy II Fe uptake-related genes [22,23,24]. Unlike the above bHLH TFs, which act as positive regulators, rice OsIRO3 and *Arabidopsis* PYE, belonging to the IVb clade bHLH, act as negative regulators of some Fe homeostasis-related genes [25,26,27,28,29]. The second module, composed of the IVb and IVc clade bHLH TFs, acts upstream of the first module. It has been demonstrated that three rice clades, IVc bHLH TFs [POSITIVE REGULATOR OF IRON HOMEOSTASIS 1 (OsPRI1)/OsbHLH060, OsPRI2/bHLH058, OsPRI3/OsbHLH059] and *Arabidopsis* IVc clade bHLH TFs [IRON DEFICIENCY TOLERANT 1 (IDT1)/bHLH034, IAA-LEUCINE RESISTANT 3(ILR3)/bHLH105,/bHLH104, and bHLH115], play a positive role in the regulation of Fe deficiency responses by directly regulating Ib gene expression [30,31,32,33,34,35,36]. Previous studies have suggested that clade IVc bHLH TF activities are regulated by post-transcriptional regulation. HEMERYTHRIN MOTIF-CONTAINING REALLY INTERESTING NEW GENE (RING)- AND ZINC-FINGER PROTEIN 1 (OsHRZ1) in rice and their ortholog BRUTUS (BTS) in *Arabidopsis*, which are the putative rice Fe sensors, can degrade OsPRI1/OsbHLH060, OsPRI2/bHLH058, and OsPRI3/OsbHLH059 and (ILR3)/bHLH105 and bHLH115, respectively [30,32,37,38]. OsbHLH061, another member of clade IVb bHLH, has been proven to be a negative regulator of Fe homeostasis by interacting with OsPRI1 and recruiting TOPLESS/TOPLESS-RELATED (TPL/TPR) repressors [39]. OsIRO3 can also inhibit the transcriptional activity of OsPRI1 by recruiting OsTPL/TPRs [29]. In *Arabidopsis*, clade IVb bHLH TF bHLH011 also negatively regulates Fe homeostasis by recruiting TPL/TPRs to inhibit clade IVc bHLH TFs [40]. Another clade IVb bHLH TF, bHLH121, forms a complex with clade IVc bHLH TFs to positively regulate Fe homeostasis [41,42,43]. Thus, the regulatory framework of Fe homeostasis in plants has been proposed to comprise HRZ/BTS → IVc/IVb bHLHs → Ib/IVb bHLHs → Fe uptake and transport-related genes. OsbHLH057 is the fourth member of the rice clade IVc bHLH TFs [25]. Recently, functional analysis of OsbHLH057 suggested that it positively regulates disease resistance and drought tolerance [44]. However, to date, there is still a lack of information on the role of OsbHLH057 in the regulation of Fe homeostasis. Therefore, the aim of the present study was to determine whether and how OsbHLH057 is involved in regulating Fe homeostasis.

Here, using a reverse genetic method, we demonstrate that OsbHLH057 is a critical factor that helps maintain Fe homeostasis. We showed that *OsbHLH057′s* transcript abundance is not changed by Fe deficiency in the roots but is induced in the shoots early in Fe deficiency. OsbHLH057 is a transcription activator localised in the nucleus. The knockout of *OsbHLH057* decreased Fe deficiency tolerance, shoot Fe concentration, and some Fe homeostasis-related gene expression. Furthermore, overexpression of *OsbHLH057* increased the shoot Fe concentration and the expression of several Fe homeostasis-related genes under Fe-sufficient conditions.

## 2. Results

### 2.1. Expression Pattern of OsbHLH057

Sequence analysis revealed that the genomic DNA of *OsbHLH057* was composed of 3126 base pairs (bp), containing five exons and four introns (Appendix A). The protein encoded by *OsbHLH057* with 256 amino acids has a typical bHLH-ZIP domain and one OsHRZ-interacting domain identified by Peng et al. (2022) [45] in the C-terminal (Appendix A). Therefore, we examined whether OsbHLH057 binds to OsHRZ1 and OsHRZ2 by use of a yeast-two-hybrid assay. The results of this assay showed that OsbHLH057 interacts with the C-terminal segments of OsHRZ1 and OsHRZ2 (Appendix A), which is consistent with previous studies [45]. Moreover, a split luciferase complementation imaging assay showed that OsbHLH057 interacts with OsHRZ1 and OsHRZ2 but not with the negative control (Appendix A), suggesting that the interaction between OsbHLH057 and OsHRZ1 or OsHRZ2 takes place in planta.

*OsbHLH057* expression was investigated in different tissues at different growth stages. The results of reverse transcription quantitative PCR (RT-qPCR) showed that *OsbHLH057* was ubiquitously expressed and primarily expressed in the leaf blades and leaf sheaths at all growth stages (Figure 1A). To further investigate whether the expression of *OsbHLH057* was affected by Fe availability, we exposed rice plants to Fe deficiency over 7 d and subsequently resupplied them with Fe for 3 d. The expression of *OsbHLH057* was unaffected by Fe deficiency in the roots (Figure 1B) and was only slightly induced in the shoots by Fe deficiency at 1 d (Figure 1C).

To further investigate the expression pattern of *OsbHLH057* in different tissues, a 2175-bp promoter of *OsbHLH057* was used to drive beta-glucuronidase (GUS) expression in rice plants. GUS staining results suggested that *OsbHLH057* was mainly expressed in the meristematic zone of root tips and the stele of the basal mature zone of the root and was highly expressed in the lateral roots (Figure 1D). In the shoot, *OsbHLH057* was highly expressed in leaf blades and sheaths but had no expression in leaf ligule and auricle (Figure 1D). In the leaf blades, *OsbHLH057* was highly expressed in vascular tissues and mesophyll cells (Figure 1D). Furthermore, the transverse section of the basal node showed that the vascular bundles had strong *OsbHLH057* expression (Figure 1D).

### 2.2. OsbHLH057 Is a Nucleus-Localised Transcription Activator

To explore the subcellular localisation of OsbHLH057, the *35S::OsbHLH057*-*GFP* vector was transiently co-transformed into rice mesophyll protoplasts with the *35S::NLS*-*mCherry* vector. The GFP signal of the OsbHLH057-GFP fusion protein was detected only in the nucleus and overlapped with the mCherry signal of the NLS-mCherry protein, a nuclear marker (Figure 2A). This observation suggests that OsbHLH057 is a nucleus-localised protein, which agrees with the predicted function of OsbHLH057 as a TF. To further examine the transcription activity of OsbHLH057, a dual luciferase reporter assay was performed. In this system, the firefly luciferase (LUC) under the control of five repeats of the GAL4 binding *cis*-element with mini 35S was used as the reporter construct, in which Renilla luciferase (REN) driven by a constitutive cauliflower mosaic virus (CaMV) 35S promoter was used as an internal control. An effector plasmid was constructed by fusing OsbHLH057 to the GAL4 DNA binding domain (BD), which was driven by the CaMV 35S promoter, and the blank vector *35S::BD* was used as the control effector (Figure 2B). As expected, the fusion protein BD-OsbHLH057-transfected tobacco leaves had a much higher value of LUC to REN than the control effector, indicating that OsbHLH057 had transcription activation activity. These results suggest that OsbHLH057 is a nucleus-localised transcription activator.

### 2.3. Loss-of-Function Mutation of OsbHLH057 Results in Decreased Fe Accumulation in the Shoot

To assess the function of OsbHLH057 in the regulation of Fe homeostasis in rice, two independent OsbHLH057 mutants were obtained using CRISPR-Cas9 technology. Two *OsbHLH057* gene sequences in the first and second exons were selected as mutation sites (Appendix A). The two homozygous *osbhlh057* mutants (*osbhlh057*-*1* and *osbhlh057*-*2*) were identified by sequencing. *osbhlh057*-*1* and *osbhlh057*-*2* had a deletion of ‘C’ and an insertion of ‘A’, respectively (Appendix A), both of which resulted in the appearance of a frameshift mutation and a premature stop codon (Appendix A). Then, we compared the growth performance of wild type (WT) and *osbhlh057* mutants under both Fe-sufficient and -deficient conditions. Regardless of Fe availability, the growth performance of the *osbhlh057* mutants was apparently poorer than that of the WT, which was consistent with the lower biomass of roots and shoots in the *osbhlh057* mutants compared with the WT (Figure 3A,D,E). After one week of Fe deficiency treatment, the new leaves from plants under Fe-deficient conditions showed chlorosis, a typical phenotype of Fe deficiency. The new leaves of the *osbhlh057* mutants had more chlorosis than the WT (Figure 3B). Consistent with this observation, the leaf soil and plant analyser development (SPAD) values of the *osbhlh057*-*1* and *osbhlh057*-*2* mutants were significantly lower than those of the WT (Figure 3C).

Furthermore, we measured the Fe concentration in the *osbhlh057* mutants. Under both Fe-sufficient and -deficient conditions, the root Fe concentrations of the *osbhlh057* mutants were similar to that of the WT (Figure 3F). The shoot Fe concentrations in the *osbhlh057* mutants were significantly lower than that in the WT control, regardless of external Fe availability (Figure 3G).

### 2.4. Overexpressing OsbHLH057 Enhances Fe Accumulation in the Shoot under Fe-Sufficient Conditions

To further clarify the functions of OsbHLH057 in the regulation of Fe homeostasis, we generated *OsbHLH057* overexpression transgenic plants containing the full-length coding sequence of *OsbHLH057* driven by a CaMV 35S promoter. RT-qPCR analysis indicated that the transcript abundance of *OsbHLH057* was significantly higher in the two independent overexpression plants (*OsbHLH057*-OE-2 and *OsbHLH057*-OE-4) than in the WT control (Appendix A). Compared with the WT plants, both OsbHLH057 overexpressing lines displayed shorter roots and shoots with less biomass under Fe-sufficient and -deficient conditions (Figure 4A,D,E). Under Fe-sufficient conditions, the Fe concentration in the shoots of *OsbHLH057*-OE-2 and *OsbHLH057*-OE-4 was higher than that of the WT, as expected (Figure 4G). Additionally, the root Fe concentration was significantly lower in *OsbHLH057*-OE-2 than in the WT and decreased in *OsbHLH05*-OE-4 compared to the WT, but there was no statistical difference (*p*-value = 0.0739) (Figure 4F). Under Fe-deficient conditions, the new leaves of *OsbHLH057*-OE-2 and *OsbHLH057*-OE-4 plants showed more chlorosis, and lower SPAD values than the WT. The shoot Fe concentrations of the *OsbHLH057*-OE-2 and *OsbHLH057*-OE-4 plants were lower than that of the WT, but there was no statistical difference between the WT and *OsbHLH057*-OE-4 plants, except for those that were unaccepted (Figure 3A–C,G). In addition, there was no substantial alteration in the root Fe concentration between the WT and *OsbHLH057*-overexpressing lines under Fe-deficient conditions (Figure 4G). These observations indicate that *OsbHLH057* overexpression results in Fe overaccumulation in the shoot under Fe-sufficient conditions but causes hypersensitivity to Fe deficiency.

### 2.5. Expression of Fe Homeostasis-Related Genes in the WT and OsbHLH057 Knockout Mutant or Overexpression Lines

Based on these results, OsbHLH057, as a transcription factor, was expected to maintain Fe homeostasis by controlling the expression of many Fe homeostasis-related genes. Therefore, we examined gene expression changes in the WT and *osbhlh057* mutant plants. In rice, *OsNAS1*, *OsNAS2*, *TOM1*, *OsYSL15*, and *OsNATT1* are representative Fe-deficiency response genes involved in Fe uptake and transport. In our RT-qPCR assay, the expression of these genes was strongly induced by Fe deficiency, as expected, and showed similar expression levels in Fe-deficient roots of WT and *osbhlh057* mutant plants under Fe-deficient conditions (Figure 5). However, the transcript abundances of *OsNAS1*, *OsNAS2*, *TOM1*, *OsYSL15*, and *OsNATT1* in the *osbhlh057* mutant were significantly repressed compared with the WT under Fe-sufficient conditions (Figure 5). The rice bHLH protein OsIRO2 is a vital regulator of Fe homeostasis that controls the expression of *OsNAS1*, *OsNAS2*, *TOM1*, *OsYSL15*, and *OsNATT1*. Therefore, we quantified the expression level of *OsIRO2* in the WT and *osbhlh057*-*1*. Similar to *OsNAS1*, *OsNAS2*, *TOM1*, *OsYSL15*, and *OsNATT1*, the *OsIRO2* expression level in *osbhlh057*-*1* was lower than in the WT under Fe-sufficient conditions but identical to the WT under Fe-deficient conditions (Figure 5).

We further investigated the gene expression changes of *OsNAS1*, *OsNAS2*, *TOM1*, *OsYSL15*, *OsNATT1*, and *OsIRO2* in *OsbHLH057*-overexpressing transgenic plants. Compared to the WT, the expression levels of these genes were strongly upregulated in the roots of the *OsbHLH057*-overexpressing lines under Fe-sufficient conditions (Figure 6). Under Fe-deficient conditions, whereas the expression levels of *OsNAS2* and *OsYSL15* were significantly downregulated in the *OsbHLH057*-overexpressing lines compared with the WT, the expression levels of *TOM1*, *OsYSL15*, *OsNATT1*, and *OsIRO2* were no different between the WT and *OsbHLH057*-overexpressing lines, except that *OsTOM1* expression was lower in the *OsbHLH057*-OE-4 than in the WT (Figure 6).

These results indicate that OsbHLH057 positively regulates the expression of many Fe homeostasis-related genes, at least under Fe-sufficient conditions.

## 3. Discussion

Plants could precisely sense and transmit Fe signals to TFs to properly regulate the expression of Fe homeostasis-related genes [30,31,32]. In rice, three clade IVc bHLH TFs, OsPRI1/OsbHLH060, OsPRI2/bHLH058, and OsPRI3/OsbHLH059, facilitate Fe homeostasis by positively regulating the expression of Fe uptake and transport-related genes. As one of the clades IVc bHLH TFs, whether OsbHLH057 is involved in regulating Fe homeostasis in rice is unknown. Determining how OsbHLH057 maintains iron homeostasis benefits a comprehensive understanding of the rice Fe homeostasis network and breeding of Fe-fortified rice. In the present study, we show that OsbHLH057 is an essential regulator in controlling Fe homeostasis.

Through RT-qPCR analysis, we found that *OsbHLH057* expression was unaffected by external Fe availability in the roots but was slightly induced by Fe deficiency and was highly expressed in the leaves (Figure 1B,C). Previous studies showed that *OsbHLH057* expression was somewhat influenced by Fe shortage in both roots and leaves [31] and was elevated in the leaves compared to the roots. The slight difference in the results of *OsbHLH057* expression in the roots under Fe-deficient conditions may be attributed to different growth conditions. For other IVc bHLH genes, Zhang et al. (2017, 2020) reported that the gene expression of *OsbHLH058*/*059*/*060* was unaffected by deficiency [30,32]. Kobayashi et al. (2019) found that *OsbHLH060* expression was slightly induced by Fe deficiency. *OsbHLH059* expression was unaffected by Fe deficiency, and *OsbHLH058* was repressed by Fe deficiency [31]. These results suggest that *IVc bHLH* genes are not easily changed by Fe deficiency at the transcript level.

Studying *osbhlh057* loss-of-function mutants revealed that OsbHLH057 is essential for the response to Fe deficiency and for maintaining Fe homeostasis (Figure 3). Compared with the WT, *OsbHLH057* knockout resulted in more chlorosis in new leaves under Fe-deficient conditions and decreased the Fe concentration in the shoots under both Fe-sufficient and -deficient conditions (Figure 3). In addition, the Fe concentration in the *OsbHLH057*-overexpressing lines further supports the idea that OsbHLH057 acts as a positive regulator of Fe homeostasis. The shoot Fe concentration in the plants overexpressing *OsbHLH057* was increased by 33.8–46.8% compared with the WT when grown under Fe-sufficient conditions (Figure 4G). Furthermore, we explored the mechanism by which OsbHLH057 positively regulates Fe homeostasis. *OsNAS1*, *OsNAS2*, *OsTOM1*, *OsYSL15*, *OsNAAT1*, and *OsIRO2* were representative of Fe uptake- and transport-related genes and Fe deficiency-induced genes. In our assay, the expression levels of *OsNAS1*, *OsNAS2*, *OsTOM1*, *OsYSL15*, *OsNAAT1*, and *OsIRO2* were strongly induced by Fe deficiency, but this process was unaffected by the *osbhlh057* mutants. However, the expression of these genes was repressed in the *osbhlh057* mutants but enhanced in the *OsbHLH057*-overexpressing lines under Fe-sufficient conditions (Figure 5 and Figure 6). Consistent with this finding, we demonstrated that OsbHLH057 exhibited transcriptional activation activity (Figure 2B,C). Moreover, *OsIRO2* expression was regulated by OsbHLH057, indicating that OsbHLH057 acts upstream of *OsIRO2*. Recently, OsbHLH057 was found to interact with OsHRZ1 and OsHRZ2 in a yeast-two-hybrid assay. We demonstrated that OsbHLH057 physically interacted with OsHRZ1 and OsHRZ2, which are situated upstream of the Fe homeostasis network, in plants and yeast (Appendix A). These data suggest that OsbHLH057 acts upstream of the Fe homeostasis network. Therefore, OsbHLH057 is crucial for Fe uptake and transport-related gene expression and hence facilitates Fe homeostasis under Fe-sufficient conditions.

To our surprise, the *OsbHLH057*-overexpressing lines displayed chlorotic leaves and decreased Fe concentration in the shoots when grown in low-Fe conditions; no statistical difference was observed between the WT and *OsbHLH057*-OE-4 (Figure 4). This unexpected result may be attributed to the secondary effect of *OsbHLH057* driven by the CaMV 35S promoter. First, the ubiquitous expression of the 35S promoter changes the tissue-specified expression of *OsbHLH057*, which may be critical for OsbHLH057 function under Fe-deficient conditions. Secondly, bHLH34/104/105/115, *Arabidopsis* IVc bHLH TFs, can form homo-or heterodimers, affecting their regulation activity [34,35,36]. Therefore, rice IVc bHLH TFs may also form homo- or heterodimers, and the high expression level of OsbHLH057 may disorder the balance of the dimerization process, especially under Fe-deficient conditions where OsbHLH057 is likely not easily degraded by OsHRZ1/2. Another explanation for the susceptibility to Fe deficiency in *OsbHLH057*-overexpressing lines may be attributed to possessing some targets that differ from their paralogs. In *Arabidopsis*, the overexpression of bHLH105 (ILR3) also showed chlorotic leaves, which may contribute to the downregulation of genes encoding chloroplast proteins, and *At*-*NEET*, which functions as a Fe-S/Fe donor in chloroplasts [46]. Thus, OsbHLH057 may indirectly downregulate some genes encoding chloroplast-related proteins to influence Fe uptake and transport under Fe-deficient conditions.

Our data indicate that the effects of OsbHLH057 on regulating Fe uptake and transport are not entirely similar to their paralogs. Previous studies have shown that all mutants of *OsPRI1/OsbHLH060*, *OsPRI2/bHLH058*, and *OsPRI3/OsbHLH059* accumulate higher levels of Fe in the roots but lower levels in the shoots, suggesting that Fe translocation from root to shoot is impaired [30,32]. For the *osbhlh057* mutants, although the Fe concentration in the roots was not significantly different, the Fe concentration in the shoots was considerably lower than in the WT (Figure 3F,G). This result supports that the translocation of Fe from root to shoot is disrupted in the *osbhlh057* mutants, which is further supported by the higher Fe concentration in the shoots but lower Fe concentration in the roots of the *OsbHLH057*-overexpressing lines under Fe-sufficient conditions (Figure 4F,G). Although there was no significant difference (*p*-value = 0.0739) based on statistical analysis, the root Fe concentration in *OsbHLH057*-OE-4 decreased 17% compared to the WT. The expression of *OsbHLH057* in the roots was mainly expressed in the stele (Figure 1D), which is also related to the function of OsbHLH057 in regulating the translocation of Fe to the shoot. Unlike the lower Fe concentrations in the roots of the *OsbHLH057*-overexpressing lines, the *OsPRI2/bHLH058*- and *OsPRI3/OsbHLH059*-overexpressing transgenic plants contained higher Fe concentrations in both the roots and shoots [32]. The differences among the lines overexpressing *OsbHLH057*, *OsPRI2/bHLH058*, and *OsPRI3/OsbHLH059* suggests that their effects on regulating Fe uptake and transport have some nuance. A slight difference also appeared in the regulation of Fe homeostasis-related genes compared with the WT; the expression of Fe homeostasis-related genes was repressed in the *osbhlh057* mutants only under Fe-sufficient conditions (Figure 5) but repressed in the mutants of *OsPRI1/OsbHLH060*, *OsPRI2/bHLH058*, and *OsPRI3/OsbHLH059* under both conditions [30,32].

In conclusion, we developed a schematic function model for OsbHLH057 (Figure 7). Under Fe-sufficient conditions, loss-of-function or overexpression of *OsbHLH057* did not affect the SPAD value but decreased or increased the shoot Fe concentration and the expression of Fe homeostasis-related genes; overexpression of *OsbHLH057* led to a slight decrease in the root Fe concentration. Under Fe-deficient conditions, both loss-of-function and overexpression of *OsbHLH057* resulted in a decreased SPAD value and shoot Fe concentration but no change in root Fe concentration and gene expression compared with the WT.

## 4. Materials and Methods

### 4.1. Plant Materials and Growth Conditions

Using a CRISPR-Cas9 genome editing system [47], we generated *OsbHLH057*-knockout mutants (*osbhlh057*-*1* and *osbhlh057*-*2*) in the *Oryza sativa* cv. *Nipponbare* background. Guide RNA (gRNA1) and gRNA2 sequences were selected and ligated into the *SK-gRNA* vector, and then the *SK-gRNA1* and *SK-gRNA2* vectors were cut with *Kpn* I and *Bgl* II. Finally, the segments containing gRNA1 and gRNA2 were ligated into the *pC1300-Cas9* vector cut with *Kpn* I and *BamH* I, respectively. To test the role of OsbHLH057, we created two *OsbHLH057*-overexpressing lines in the Nipponbare background in which two independent lines (*OsbHLH057*-OE-2 and *OsbHLH057*-OE-4) were used for the subsequent analysis. For the construction of the *OsbHLH057*-overexpressing vector, the full-length CDS of *OsbHLH057* was cloned from the cDNA of Nipponbare and first inserted into the vector *pDONR221* and finally recombined into the vector *pGWB2* [48]. To construct the *ProOsbHLH057::GUS* vector, a 2175-bp genomic DNA was amplified by 2×Hieff Canace^®^ Plus PCR Master Mix (Yeasen, Shanghai, China) and recombined into the vector *pCAMBIA1300-GUS* using ClonExpress^®^ II One Step Cloning Kit (Vazyme, Nanjing, China). The above resultant plasmids were transferred to *Agrobacterium tumefaciens* EHA105. Transformations to the callus of Nipponbare were carried out as described previously [49]. All primers used for the construction of vectors are listed in Appendix A.

After germination in water for two days at 37 °C, the seeds were transferred to a net floating on 0.5 mM CaCl_2_ solution and kept dark for three days. At the fourth day, CaCl_2_ solution was replaced by one-half-strength Kimura B solution which containing 0.18 mM (NH_4_)_2_SO_4_, 0.27 mM MgSO_4_, 0.09 mM KNO_3_, 0.18 mM CaNO_3_, 0.09 mM KH_2_PO_4_, 0.50 μM MnCl_2_, 3.00 μM H_3_BO_3_, 1.00 μM (NH_4_)_6_Mo_7_O_2_, 0.40 μM ZnSO_4_, 0.20 CuSO_4_, and 2.00 μM FeSO_4_. The solution pH was adjusted to 5.5 and renewed every two days. Seedlings were grown in a greenhouse with 14 h 30 °C: 10 h 25 °C, light: dark cycles. For Fe deficiency treatments, the FeSO_4_ was removed from the solution.

### 4.2. RNA Isolation and RT-qPCR

To examine the tissue-specific expression of *OsbHLH057* in different growth stages, different tissues, including root, basal node, leaf blade, leaf sheath, node I, panicle, and seed, from rice plants at vegetative growth, flowering, or grain filling were sampled, as described previously [50]. To investigate the Fe deficiency response of *OsbHLH057*, two-week-old seedlings were treated without Fe for seven days (7 d) and then resupplied with 2 μM FeSO_4_ for 3 d. The roots and shoots were collected at 1, 3, 5, and 7 d in Fe deficiency treatment and 1 and 3 d in the Fe resupply stage. For examining the relative expression of *OsbHLH057* in the *OsbHLH057*-overexpressing lines, roots of the wild type (WT) and *OsbHLH057*-overexpressing lines cultivating in the solution with 2 μM FeSO_4_ was collected and stored at −80 °C 

For analyzing gene expression influenced by OsbHLH057, wild type, *OsbHLH057*-knockout mutants, and *OsbHLH057*-overexpressing lines were planted in solution with or without Fe for one week, and the roots were collected for RNA extraction.

Total RNA was extracted using TaKaRa Universal RNA Extraction Kit (TaKaRa, Dalian, China) and then synthesized to cDNA using TaKaRaPrimeScript™ 1st Strand cDNA Synthesis Kit (TaKaRa, Dalian, China). The subsequent cDNA was used for real-time quantitative PCR (RT-qPCR) using ChamQ™ SYBR^®^ Color qPCR Master Mix (Vazyme, Nanjing, China) on a Mastercycler^®^ ep realplex real-time PCR system (Eppendorf, Hamburg, Germany). *OsActin1* was amplified as an internal control. The relative gene expression level was calculated by the Equation 2^−△△Ct^. All primers used for RT-qPCR are listed in Appendix A.

### 4.3. Subcellular Localization Analysis

For subcellular localization, the coding sequence (CDS) without stop codon of *OsbHLH057* was cloned into the N terminus of GFP in the *pYL322-d1-eGFP* vector using ClonExpress^®^ II One Step Cloning Kit (Vazyme, Nanjing, China) to generate *35S::OsbHLH057-GFP* vector. The *35S::NLS-mCherry* vector was used as a nuclear marker. The co-transformed *35S::OsbHLH057-GFP* and *335S::NLS-mCherry* vectors were transiently expressed in rice protoplasts, as described previously [51]. As a negative control, *35S::GFP* and *335S::NLS-mCherry* vectors were also co-transformed. The fluorescence signals were observed using a laser confocal microscope (UltraVIEW VOX, PerkinElmer, Waltham, MA, USA). Primers used for subcellular localization are listed in Appendix A.

### 4.4. Transcription Activity Analysis

For the GAL4-dependent chimeric transactivation assay, transient dual-luciferase expression assays were performed. The full-length CDS of *OsbHLH057* was amplified and fused into the effector vector *pCAMBIA1300-BD,* creating *35S::BD*-*OsbHLH057*. The reporter plasmid *5×GAL4-mini35S::firefly luciferase* (*LUC*) containing *35S::renilla firefly luciferase* (*REN*) internal control was used before [39]. Combinations of these effector vectors (*35S::BD*-*OsbHLH057* and *35S::BD*) and reporter vectors were transformed into tobacco leaves using *Agrobacterium*-mediated transformation, as described previously [39]. In *Agrobacterium*-mediated transformation, the bacteria expressing the corresponding vector were cultured, harvested, and re-suspended in Murashige and Skoog-MES medium containing 10 mM MES, 0.2 mM acetosyringone, and 10 mM MgCl_2_ (pH = 5.6) to the highest concentration of OD600 = 0.5. Then, the *Agrobacterium* were mixed and infiltrated into *N. benthamiana* leaves. The infiltrated leaves were sampled for measuring LUC and REN activities using a Dual-Luciferase Reporter Assay Kit (Yeasen, Shanghai, China). The activity of LUC to REN under BD control was set to 1. The primers used for transcription activation assays are given in Appendix A.

### 4.5. Yeast-Two-Hybrid Assay

For the yeast-two-hybrid-assay, the full-length CDS of OsbHLH057 was cloned and fused into the vector *pGADT7* (*AD*) to generate *AD-OsbHLH057*. The C terminus of OsHRZ1 and OsHRZ2 were inserted into the vector *pGBKT7* (*BD*) to form BD- OsHRZ1C and BD-OsHRZ2C, respectively. These *AD* and *BD* vectors were transformed into AH109 cells. After culturing on the synthetic dropout nutrient medium lacking tryptophan and leucine and the synthetic dropout nutrient medium lacking tryptophan, leucine, histidine, and adenine plates at 30 °C for 2 d. The yeast cells could grow on both selective mediums, which indicated protein–protein interactions. The primers used for the yeast-two-hybrid-assay are listed in Appendix A.

### 4.6. Split-LUC Complementation Assay

The C terminus of *OsHRZ1* or *OsHRZ2* without a stop codon was amplified from rice cDNA and inserted into *pCAMBIA1300-nLUC* [52] vector, and the full-length coding sequence of OsbHLH057 was amplified and fused with cLUC in the vector of *pCAMBIA1300-cLUC* [52] through homologous recombination using the ClonExpress II One Step Cloning Kit (Vazyme, Nanjing, China). Combination proteins, OsHRZ1C-nLUC or OsHRZ1C-nLUC and cLUC-OsbHLH057, cLUC-OsbHLH057 and nLUC, OsHRZ1C-nLUC or OsHRZ1C-nLUC and nLUC, and cLUC and nLUC were transformed into tobacco leaves via *Agrobacterium*-mediated transformation, as described above transcription activity assay. The transinfected leaves were sampled for LUC signal detection using a Tanon 5200 Multi automatic chemiluminescence/fluorescence image analyzer (Tanon, Shanghai, China). The reconstruction of the LUC signal indicated the occurrence of interaction between proteins. The primers used for the construction of vectors are listed in Appendix A.

### 4.7. Measurement of SPAD Values and Fe Concentrations

12-day-old seedlings of the WT and *OsbHLH057*-knockout and -overexpressing were transferred in nutrient solution containing 2 or 0 μM FeSO_4_ and grown for 7 d. The portable chlorophyll meter (SPAD-502; Konica Minolta Sensing, Osaka, Japan) was used to measure the SPAD values of the new fully expanded leaves. The roots and shoots were collected for Fe concentration analysis. The method used to digest the roots and shoots was according to Dong et al., 2018 [53]. The Fe concentration was examined using inductively coupled plasma mass spectrometry (ICP-MS; NexION 300X; Perkin-Elmer, Waltham, MA, USA).

### 4.8. Histochemical GUS Staining

Histochemical GUS staining was performed in the Pro*OsbHLH057*::*GUS* transgenic rice plants. Various organs of seedlings grown in nutrient solution containing 2 μM FeSO_4_ were harvested and subjected to GUS staining as described previously [24]. After vacuum treatment for 30 min, the samples were incubated at 37 °C overnight and decolorized with 95% ethanol. Photographs were taken with a stereo microscope (Nikon, Tokyo, Japan). Sections of 20 µm thickness were cut and photographed using a vibratome (VT1200S, Leica, Nussloch, Germany) and a microscope (DM500, Leica, Nussloch, Germany), respectively.

### 4.9. Statistical Analysis

Data analysis was performed using SPSS v.20.0 (IBM Corp, Armonk, NY, USA). Data were shown as means ± SD. Differences in the means between two groups were compared using a two-tailed Student’s *t*-test and among three or more groups using one-way ANOVA followed by Duncan’s multiple-range test.

### 4.10. Accession Numbers

Sequence data from this article can be found in the Rice Genome Annotation Project database under the following accession number: *OsbHLH057* (LOC_Os07g35870), *OsPRI1* (LOC_Os08g04390), *OsPRI2* (LOC_Os05g38140), *OsPRI3* (LOC_Os02g02480), *OsIRO2* (LOC_Os01g72370), *OsNAS1* (LOC_Os03g19427), *OsNAS2* (LOC_Os03g19420), *OsNAAT1* (LOC_Os02g20360), *OsTOM1* (LOC_Os11g04020).

## Figures and Tables

**Figure 1 ijms-23-14869-f001:**
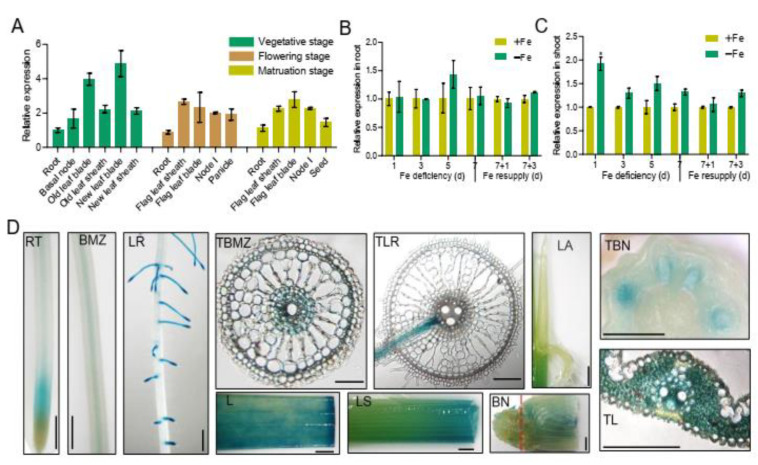
Expression profiling of *OsbHLH057* in rice. (**A**) Reverse transcription quantitative PCR (RT-qPCR) analysis of the expression levels of *OsbHLH057* in various rice tissues of different growth stage. (**B**,**C**) RT-qPCR analysis of the expression levels of *OsbHLH057* in the root (**B**) and shoot (**C**) under different Fe supply conditions. (**D**) GUS staining of various tissues in *ProOsbHLH057::GUS* transgenic plants, including root tips (RT), basal mature zone of root (BMZ), lateral roots (LR), transverse section of BMZ (TBMZ), transverse section of TR (TLR), leaf (L), leaf sheaths (LS), ligule and auricle (LA), basal node (BN), transverse section of BN (TBN), and transverse section of L (TL). Red line in BN indicates the place of section. Scale bars = 50 μM in TBMZ, TLR, and TL; Scale bars = 1 mm in other pictures.

**Figure 2 ijms-23-14869-f002:**
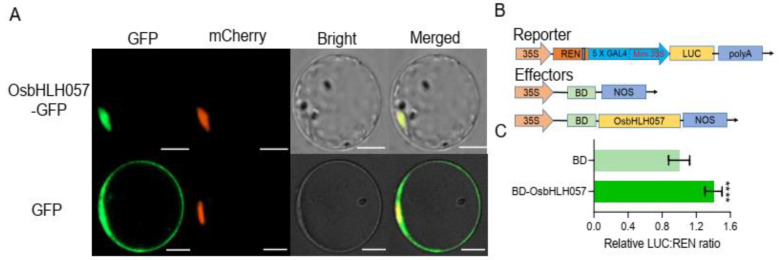
Subcellular localization and transcriptional activity of OsbHLH057. (**A**) Subcellular localization of OsbHLH057 in rice protoplasts. OsbHLH057 was fused with GFP to obtain the OsbHLH057-GFP fusion protein. Nucleus localization signal (NLS) fused with mCherry was used as a nuclear marker. The upper row indicates the fluorescent signal from co-expressing OsbHLH057-GFP and NLS-mCherry. As a negative control, the bottom row shows the fluorescent signal from co-expressing GFP and NLS-mCherry. Scale bars, 8 μm. (**B**) Diagram of vectors of the OsbHLH057 transcriptional activity assay. Five repeats of GAL4 binding *cis*-element with mini 35S were fused with firefly luciferase (LUC) as the reporter vector in which 35S driven Renilla luciferase (REN) as an internal control. The full-length coding sequence of *OsbHLH057* was fused with the GAL4 DNA binding domain (BD) to create *35S::BD-OsbHLH057* effector vector. The empty vector *35S::BD* was used as a control. (**C**) OsbHLH057 has transcription activation activity. The reporter plasmid was co-transformed with *35S::BD* or *35S::BD-OsbHLH057* into tobacco leaves using *Agrobacterium*-mediated transformation. The relative LUC:Ren ratio under BD was set to 1. Values are means ± SD of six technical replicates. Asterisks indicate significant differences between BD and OsbHLH057 based on two-tailed Student’s *t*-test (**** *p* < 0.0001).

**Figure 3 ijms-23-14869-f003:**
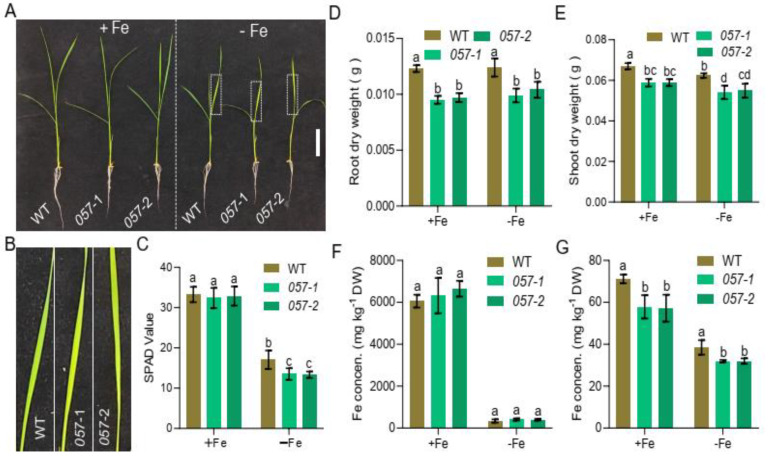
Phenotypes of *OsbHLH057* loss-off-function mutants. 12-day-old seedlings of WT, *osbhlh057-1*, and *osbhlh057-2* were shifted in nutrient solution with or without Fe for 7 d. (**A**) Pictures of 19-d-old seedlings. Scar bars, 5 cm. (**B**) New leaves from seedlings under Fe-deficient condition. Magnification of part leaves in the dotted line in (**A**) with 5 times. (**C**) The SPAD of the third leaves. (**D**,**E**) Root and shoot biomass. Root (**D**) and shoot (**E**) dry weight. (**F**,**G**) Fe concentration in the WT, *osbhlh057-1*, and *osbhlh057-2.* Root (**F**) and shoot (**G**) Fe concentration. Data in (**C**–**E**) and (**F**,**G**) represent the means ± standard deviation (SD) of six and three biological replicates, respectively. Means with different letters are statistically significant differences as determined by one-way ANOVA followed by Duncan’s multiple-range test (*p* < 0.05).

**Figure 4 ijms-23-14869-f004:**
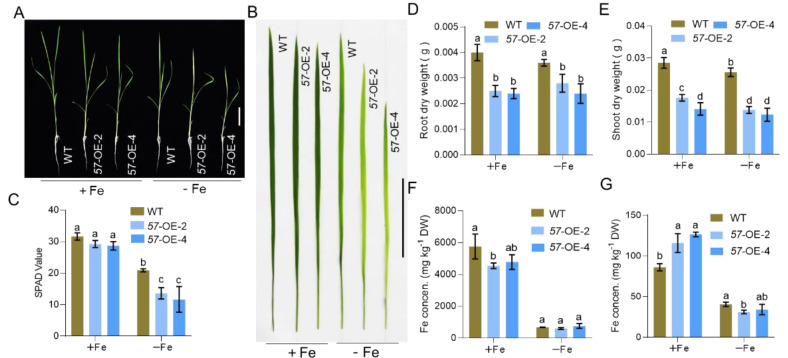
Phenotypes of *OsbHLH057* overexpression lines. (**A**) Images of wild-type (WT) and *OsbHLH057* overexpression lines. Bar, 5 cm. (**B**) New leaves of WT and *OsbHLH057* overexpression lines. Bar, 5 cm. (**C**) Soil and plant analyzer development (SPAD) values of new leaves of WT and *OsbHLH057* overexpression lines. Data represent the means ± SD of six biological replicates. (**D**,**E**) Root length (**D**) and shoot height (**E**) of WT and *OsbHLH057* overexpression lines. Data represent the means ± SD of six biological replicates. (**F**,**G**) Fe concentrations in the root (**F**) and shoot (**G**) of WT and *OsbHLH057* overexpression lines. Data represent the means SD of four biological replicates. Here, 12-d-old rice seedlings of WT and *OsbHLH057* overexpression lines were transferred to a solution containing 0 (Fe) or 2 μM (+Fe) FeSO_4_ and grown for 8 d. Means with different letters indicate statistically significant differences as determined by one-way ANOVA followed by Duncan’s multiple-range test (*p* < 0.05).

**Figure 5 ijms-23-14869-f005:**
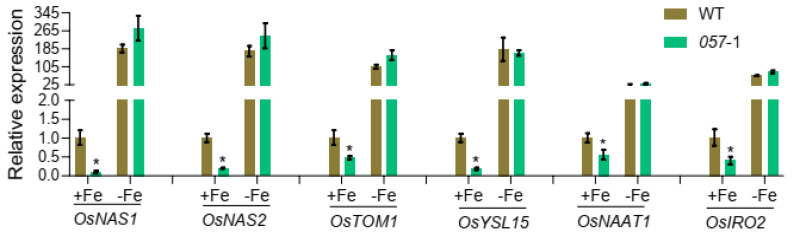
Expression of Fe homeostasis-related genes in the WT and *osbhlh057-1* mutant. 12-day-old seedlings of the WT and *osbhlh057-1* mutant were transferred to 1/2 Kimura B solution containing 2 μM FeSO_4_ or 0 μM FeSO_4_ for 4 days. The roots were sampled for gene expression analysis. *OsActin1* was used as an internal control. Data are means ± SD of three biological replicates. Asterisks indicate significant differences between WT and *osbhlh057-1* based on two-tailed Student’s *t*-test (* *p* < 0.05).

**Figure 6 ijms-23-14869-f006:**
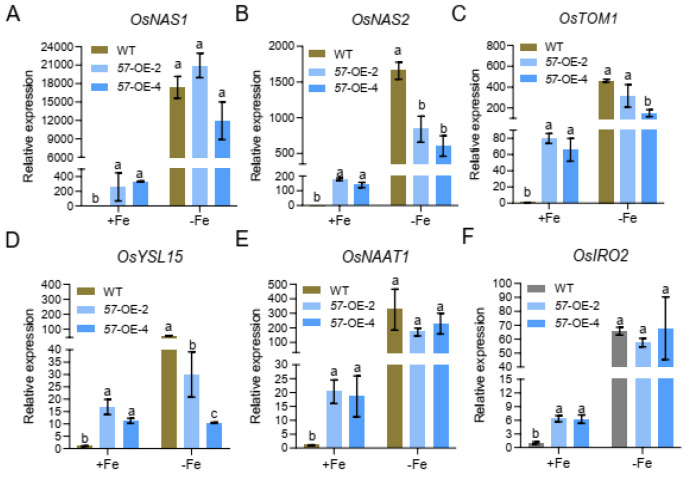
Expression of Fe homeostasis-related genes in the WT and *OsbHLH057* overexpression lines. 12-day-old the WT and *OsbHLH057* overexpression seedlings were transferred to nutrient solution with 2 μM FeSO_4_ or without FeSO_4_ for 7 days. The expression of *OsNAS1* (**A**), *OsNAS2* (**B**), *OsTOM1* (**C**), *OsYSL15* (**D**), *OsNAAT1* (**E**), and *OsIRO2* (**F**) in the roots were analyzed. *OsActin1* was used as an internal control. Data are means ± SD of three biological replicates. Means with different letters indicate statistically significant differences as determined by one-way ANOVA followed by Duncan’s multiple-range test (*p* < 0.05).

**Figure 7 ijms-23-14869-f007:**
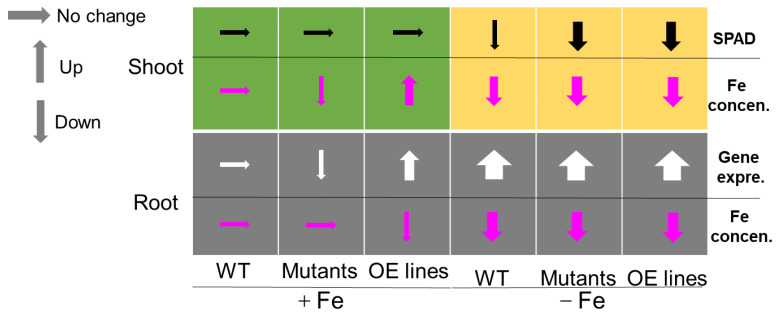
A schematic of OsbHLH057 in regulating Fe homeostasis. The effect of loss-of-function and overexpression of *OsbHLH057* on increasing (up arrows) or decreasing (down arrows) physiological processes in the roots and shoots under Fe-sufficient (+Fe) and Fe-deficient (−Fe) conditions. Black arrows, SPAD value; pink, Fe concentration; white, gene expression.

## Data Availability

Data are contained within the article or Appendix A.

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
