# Peer review of "The bHLH Transcription Factor OsbHLH057 Regulates Iron Homeostasis in Rice"

_ijms, 2022, doi:10.3390/ijms232314869_

Round 1

Reviewer 1 Report

Dear authors,

Please consider the following comments to improve the manuscript.

-Introduction should be rewritten by detailing the aim and concept of the study. The abstract should state briefly the purpose of the study, the principal results and major conclusions.

- Introduction is very general and need to be elaborative to explore the actual philosophy to design the experiment. The introduction is insufficient to provide the state of the art in the topic. The originality and novelty of the paper need to be further clarified. What progress against the most recent state-of-the-art similar studies was made in this study?   

- Materials and Methods part should be after the introduction.

-Justify the novelty in discussion.

-Discuss the statistical analysis section properly.

-Check and correct grammatical errors throughout the article.

Author Response

-Introduction should be rewritten by detailing the aim and concept of the study.

Response: We have rewritten the introduction as suggested. The new version of introduction meets the aim and concept of the study as specification.

The abstract should state briefly the purpose of the study, the principal results and major conclusions.

Response: We have clearly and briefly pointed out the purpose of the study, the principal results, and major conclusions in the new version of the abstract.

- Introduction is very general and need to be elaborative to explore the actual philosophy to design the experiment. The introduction is insufficient to provide the state of the art in the topic. The originality and novelty of the paper need to be further clarified. What progress against the most recent state-of-the-art similar studies was made in this study?

Response: The review clearly pointed out shortcomings of our old version of introduction. We fully agree with the existence of these shortcomings pointed out by the review. Therefore, we have rewritten a significantly portion of the introduction as suggested. We showed more detailed background of bHLH TFs in regulating Fe homeostasis in the rice and Arabidopsis, not only in the rice as before. Furthermore, we present the aim of this study was to determine whether and how OsbHLH057 is involved in regulating Fe homeostasis because there is still a lack of information on the role of OsbHLH057 in the regulation of Fe homeostasis. We consider that disclosing the function of OsbHLH057 is necessary, it is a homologue of some Fe homeostasis-related bHLHs though, and benefits a comprehensive understanding of the rice Fe homeostasis network.

- Materials and Methods part should be after the introduction.

Response: We have re-confirmed the IJMS format carefully. Materials and Methods part is followed by the discussion.

-Justify the novelty in discussion.

Response: We have added some content to justify the novelty. Firstly, we definitely pointed out the OsbHLH057 is a new gene involved in Fe homeostasis in the first paragraph. Secondly, we supplied the data to support that OsbHLH057 an essential role in the third paragraph. Finally, we elaborately compared the OsbHLH057 with its paralogs, and then found that OsbHLH057 has its specific points in the second, fourth, and fifth paragraph.

-Discuss the statistical analysis section properly.

Response: Thank you for the review to point out this point where the Fe concentrations in OsbHLH057-OE-4 was not lower than tant in WT.  We have re-presented the results with p-value (line222-223) in the results part. We also discussed this point in the discussion part.

-Check and correct grammatical errors throughout the article.

Response: We have check and correct grammatical errors throughout the article carefully by ourself and Professional editing service.

Reviewer 2 Report

The manuscript submitted by Wang et al entitled The bHLH Transcription Factor OsbHLH057 Regulates Iron 3 Homeostasis in Rice attempts to characterize OsbHLH057 and to highlight its implication in Fe homeostasis. One can follow the results and the text is readable. My only concern is that the crucial role of this transcription factor is neither supported by the results nor comprehensively discussed. The authors might consider revising the introduction and the discussion accordingly, and provide a schematic representation of their conclusions (shoot vs root; wild type, mutants, overexpression; Fe sufficient vs Fe deficient conditions).    

Other comments:

- Line 82-84 please revise the sentence.

Author Response

The manuscript submitted by Wang et al entitled “The bHLH Transcription Factor OsbHLH057 Regulates Iron 3 Homeostasis in Rice” attempts to characterize OsbHLH057 and to highlight its implication in Fe homeostasis. One can follow the results and the text is readable.

My only concern is that the “crucial” role of this transcription factor is neither supported by the results nor comprehensively discussed.

Response: In the old version of the subscript, due to the special relevance of OsbHLH057 did not be detailly presented and discussed, we may not clearly show that the crucial role of OsbHLH057 to the review. Indeed, to study the role of OsbHLH057 in regulating Fe homeostasis in rice, we created the both knockout mutants and overexpression lines of OsbHLH057. The results present by here show that both knockout and overexpressing OsbHLH057 resulted in the change of Fe concentration and many Fe-homeostasis genes expression (Figure 3-7). Moreover, we confirmed that OsbHLH057 was a direct target of OsHRZ1 and OsHRZ2 (Figure S2), which are putative Fe sensors and play a crucial role in regulating Fe homeostasis. The expression pattern and subcellular localization of OsbHLH057 were also examined. These results are enough to support that OsbHLH057 is a crucial regulator of Fe homeostasis. In addition, we also have rewritten a part of the discussion. In the discussion, we elaborately discussed why OsbHLH057 is a crucial regulator of Fe homeostasis in rice, especially in the second paragraph of the new version discussion part.

The authors might consider revising the introduction and the discussion accordingly, and provide a schematic representation of their conclusions (shoot vs root; wild type, mutants, overexpression; Fe sufficient vs Fe deficient conditions).    

Response: We have rewritten a significant portion of introduction and a part of the discussion. We have provided a schematic representation of conclusions in Figure 7.

Other comments:

- Line 82-84 please revise the sentence.

Response: We have revised the sentence in Line 82-84 of old version. In the new version, the sentence was presented in the line 91-94.

Round 2

Reviewer 1 Report

I appreciate your time and effort taking care of the suggestions which makes the article much better now.